# Genetically Predicted Type 2 Diabetes Mellitus Liability, Glycated Hemoglobin and Cardiovascular Diseases: A Wide-Angled Mendelian Randomization Study

**DOI:** 10.3390/genes12101644

**Published:** 2021-10-19

**Authors:** Bowen Liu, Amy M. Mason, Luanluan Sun, Emanuele Di Angelantonio, Dipender Gill, Stephen Burgess

**Affiliations:** 1Department of Public Health and Primary Care, University of Cambridge, Cambridge CB1 8RN, UK; bowen.liu@wolfson.ox.ac.uk (B.L.); am2609@medschl.cam.ac.uk (A.M.M.); ls811@medschl.cam.ac.uk (L.S.); ed303@medschl.cam.ac.uk (E.D.A.); 2Nuffield Department of Population Health, University of Oxford, Oxford OX3 7LF, UK; 3Department of Epidemiology and Biostatistics, School of Public Health, Imperial College London, London W2 1PG, UK; 4Genetics Department, Novo Nordisk Research Centre Oxford, Old Road Campus, Oxford OX3 7FZ, UK; 5Clinical Pharmacology and Therapeutics Section, Institute of Medical and Biomedical Education and Institute for Infection and Immunity, St George’s, University of London, London SW17 0RE, UK; 6Clinical Pharmacology Group, Pharmacy and Medicines Directorate, St George’s University Hospitals NHS Foundation Trust, London SW17 0QT, UK; 7Medical Research Council Biostatistics Unit, University of Cambridge, Cambridge CB2 0SR, UK

**Keywords:** mendelian randomization, type 2 diabetes mellitus, hemoglobin A1c, average blood glucose, cardiovascular diseases

## Abstract

(1) Aim: To investigate the causal effects of T2DM liability and glycated haemoglobin (HbA1c) levels on various cardiovascular disease outcomes, both in the general population and in non-diabetic individuals specifically. (2) Methods: We selected 243 variants as genetic instruments for T2DM liability and 536 variants for HbA1c. Linear Mendelian randomization analyses were performed to estimate the associations of genetically-predicted T2DM liability and HbA1c with 12 cardiovascular disease outcomes in 367,703 unrelated UK Biobank participants of European ancestries. We performed secondary analyses in participants without diabetes (HbA1c < 6.5% with no diagnosed diabetes), and in participants without diabetes or pre-diabetes (HbA1c < 5.7% with no diagnosed diabetes). (3) Results: Genetically-predicted T2DM liability was positively associated (*p* < 0.004, 0.05/12) with peripheral vascular disease, aortic valve stenosis, coronary artery disease, heart failure, ischaemic stroke, and any stroke. Genetically-predicted HbA1c was positively associated with coronary artery disease and any stroke. Mendelian randomization estimates generally shifted towards the null when excluding diabetic and pre-diabetic participants from analyses. (4) Conclusions: This genetic evidence supports causal effects of T2DM liability and HbA1c on a range of cardiovascular diseases, suggesting that improving glycaemic control could reduce cardiovascular risk in a general population, with greatest benefit in individuals with diabetes.

## 1. Introduction

Macrovascular disorders, such as coronary artery disease, peripheral vascular disease and stroke, are common consequences of type 2 diabetes mellitus (T2DM), and are responsible for more than 50% of deaths in such patients [1]. Glycated haemoglobin (HbA1c) is an indicator of long-term glycaemic control and has also shown positive associations with coronary artery disease and ischemic stroke risk, even in individuals without diabetes [2]. However, the impact of lowering average blood glucose levels on different cardiovascular diseases is not known, and neither are its effects in individuals without diabetes.

Mendelian randomization (MR) is an approach that can be used to assess the causal roles of T2DM liability and dysglycaemia on cardiovascular risk. By using genetic variants associated with the exposure as instrumental variables, estimates from MR are less prone to bias from environmental confounding and reverse causation than conventional observational associations. Previous Mendelian randomization studies have shown positive associations of genetic liability to T2DM with coronary artery disease (CAD) and ischemic stroke [3,4], and of genetically-predicted HbA1c levels with CAD risk in the normoglycemic range [5]. However, MR has not been used to investigate the relationships of T2DM liability and HbA1c with a wide range of cardiovascular diseases, including aortic aneurysm, peripheral vascular disease, venous thromboembolism, heart failure, and aortic valve stenosis. Additionally, the impact of dysglycaemia on cardiovascular disease potentially varies between patients with diabetes and individuals with normal or pre-diabetic blood glucose levels. Our previous MR study suggested a linear dose-response relationship between average blood glucose and risk of CAD across the normoglycemic range [5]. However, there is no MR study comparing the impact of lowering average blood glucose level on CAD or other cardiovascular diseases between diabetes-free individuals and individuals with diabetes.

In this study, we employed MR to investigate the causal effects of increased T2DM liability and elevated HbA1c on risk of 12 cardiovascular diseases in the UK Biobank study. We also performed complementary analyses excluding those with a T2DM diagnosis or elevated HbA1c levels to investigate evidence for an effect of liability to T2DM or elevated HbA1c in individuals without T2DM and with HbA1c levels in the normal range.

## 2. Methods

### 2.1. Overview of Study Design

We performed linear MR analyses to estimate the associations of genetic T2DM liability and genetically-predicted HbA1c with risk of cardiovascular diseases in UK Biobank, a population-based cohort study of UK residents aged between 40 and 69 at recruitment. Primary analyses were performed using genetic associations with 12 cardiovascular diseases estimated in 367,703 unrelated UK Biobank participants of European ancestries [6]. Outcomes were defined based on hospital episode statistics (using International Classification of Diseases 9th and 10th Edition codings), operations codes (using Office of Population Censuses and Surveys Classification of Interventions and Procedures version 4 codings), and self-reported events from an interview with a nurse practitioner (details can be found in Appendix A). We also performed secondary analyses restricted to non-diabetic individuals: excluding those with diabetes, and excluding those with diabetes or pre-diabetes. In supplementary analyses for HbA1c, we omitted variants associated with erythrocytic traits to exclude the influence of genetic variants affecting HbA1c by erythrocytic rather than glycaemic pathways. We also assessed the consistency of evidence in multivariable MR analyses adjusting for genetically-predicted body mass index (BMI) to exclude any potential causal effect acting via obesity, either as a pleiotropic effect of a genetic variant or as a mediator of the effect of HbA1c or T2DM liability.

### 2.2. Selection of Genetic Variants as Instruments

Genetic instruments for T2DM liability were selected from a genome-wide association study of the DIAGRAM consortium, which consists of 74,124 T2DM cases and 824,006 controls of European ancestries [7]. A total of 243 loci were identified as associated with T2DM at a genome-wide significance level of *p* < 5 × 10^−8^, where only the lead variants were included as instruments for T2DM liability. Genetic instruments for HbA1c were selected from a genome-wide association study in 408,989 UK Biobank participants of European ancestries and without clinically diagnosed diabetes or history of antidiabetic medications, without HbA1c > 6.5% or random glucose >200 mg/dL [4]. The study identified 543 independent variants associated with HbA1c at a genome-wide significance level of *p* < 5 × 10^−8^ (following pruning for linkage disequilibrium at r2 < 0.01); after excluding 7 complex variants with unclear allele orientation, a total of 536 variants were selected as instrumental variables for HbA1c.

### 2.3. Genetic Associations with Cardiovascular Outcomes

Genetic associations with outcomes were estimated using logistic regression with adjustment for age, sex, and 10 genomic principal components to account for population structure. The analytic sample was defined using a previously described approach [8]: we excluded participants with genetic sex mismatch, non-European ancestries (self-report or inferred by genetics), or excess heterozygosity (>3 standard deviations from the mean), and included only one of each set of related participants (third-degree relatives or closer).

### 2.4. Statistical Analyses

We used the multiplicative random-effects inverse variance weighted (IVW) method to estimate the associations of genetically-predicted T2DM liability and HbA1c respectively with each of the outcomes [9]. Estimates were scaled to a doubling in genetically-predicted T2DM risk, or to a 1% increase in genetically-predicted HbA1c. For analyses in non-diabetic individuals, we defined two subsamples: participants without diabetes (residual HbA1c <6.5% and no baseline diagnosis of T2DM) and participants without diabetes or pre-diabetes (residual HbA1c <5.7% and no baseline diagnosis of T2DM). Diagnosis of T2DM was defined at recruitment based on self-reported disease history and medical records according to a previous specified algorithm: those with either “probable” or “possible” T2DM were excluded from the analysis [10]. Residual HbA1c is defined as the residual value of HbA1c after regression on the genetic variants used as instruments, and represents the expected value HbA1c would take for that individual if their genetic variants all took their mean value. Selection on residual HbA1c rather than HbA1c avoids inducing collider bias, as residual HbA1c is independent of the genetic variants used in the analysis. We performed sensitivity analyses using the weighted median and MR-Egger methods to assess the consistency of estimates under alternative assumptions about genetic pleiotropy [9]. We also performed Cochran’s Q test to assess the heterogeneity between estimates obtained using different variants. We calculated the statistical power of primary analyses to detect a true odds ratio of 1.01, 1.02, 1.05 and 1.1 associated with 1 mmol/mol increment in genetically-predicted HbA1c for each outcome.

To exclude the impact of genetic variants affecting HbA1c levels through erythrocytic pathways, we performed supplementary analyses using HbA1c variants which were not associated with erythrocytic traits [4]. According to a previously described approach, genetic variants associated (*p*-value < 0.001) with any hematological traits related to the count, structure and function of red blood cells were excluded from this analysis [4]. A total of 213 variants were identified as not associated with any erythrocytic factors and were used in these analyses as instruments.

Finally, to account for any potential pleiotropic effect of genetic variants on BMI or an effect of the exposure mediated through BMI, multivariable MR analyses were performed for T2DM liability and HbA1c separately with BMI included as a risk factor. Genetic associations with BMI were extracted from a meta-analysis of genome-wide association studies conducted in UK Biobank and GIANT consortium, combining 649,649 participants with European ancestry [11].

Genetic associations with outcomes were estimated using snptest, and all other statistical analyses were performed using Rstudio version 1.2.5033. As 12 outcomes were assessed, a Bonferroni-corrected significance level of 0.05/12 = 0.004 was used as the threshold for statistical significance. *p*-values between 0.004 and 0.05 are described as suggestively significant.

## 3. Results

The mean age of UK Biobank participants was 57.1 years (standard deviation 8.0), and 54.1% of participants were female (Table 1). The mean HbA1c level was 35.5 mmol/mol (5.4%) in all participants. The 536 genetic variants for HbA1c explained 1.9% of the variance in HbA1c, corresponding to an F-statistic of 14.8. Power calculations are presented in Appendix A. Power to detect an odds ratio of 1.05 per 1 mmol/mol increase in genetically-predicted HbA1c was close to 100% for CAD, but only 12.7% for thoracic aortic aneurysm and 30.7% for abdominal aortic aneurysm, suggesting that power was adequate for more common outcomes, but low for less common outcomes.

In primary analyses, genetically-predicted T2DM liability was significantly associated with (ordered from largest estimate decreasing): peripheral vascular disease, aortic valve stenosis (non-rheumatic), CAD, heart failure, ischaemic stroke, and any stroke (Figure 1 and Appendix A). A suggestive association was observed for deep vein thrombosis. Associations with haemorrhagic stroke and aortic aneurysm outcomes were compatible with the null. When excluding participants with diabetes and then either diabetes or pre-diabetes, associations attenuated substantially. When excluding participants with either diabetes or pre-diabetes, none of the associations remained even at a suggestive level of significance. Estimates from sensitivity analyses using the weighted median and MR-Egger method were generally similar (Appendix A). The significant Y-intercepts of MR-Egger analyses for T2DM liability with CAD and heart failure indicated the potential directional pleiotropy biasing these analyses. Substantial heterogeneity in the variant-specific estimates was observed for several outcomes (Appendix A).

Genetically-predicted HbA1c was significantly associated with CAD and any stroke (Figure 2 and Appendix A). Suggestive associations were observed for haemorrhagic stroke, peripheral vascular disease, and pulmonary embolism. Estimates generally shifted towards the null on exclusion of diabetics, and further attenuated on the exclusion of diabetics and pre-diabetics. An exception was haemorrhagic stroke for which associations increased slightly, and were significant on exclusion of diabetics and pre-diabetics. The association with CAD risk remained significant on exclusion of diabetics, but not on exclusion of diabetics and pre-diabetics. Similar associations were observed for CAD, any stroke, and peripheral vascular disease in supplementary analyses excluding variants associated with an erythrocytic trait (Appendix A), suggesting that the positive estimates for HbA1c are driven by dysglycaemia and not other functions of HbA1c. In contrast, associations with pulmonary embolism and haemorrhagic stroke were attenuated. Point estimates obtained using the weighted median and MR-Egger methods were generally similar, but with wider confidence intervals reflecting their lower statistical power (Appendix A). As with HbA1c, substantial heterogeneity in the variant-specific estimates was observed for several outcomes (Appendix A).

Multivariable MR was performed for outcomes with significant or suggestively significant estimates in primary analyses (Appendix A). Estimates for T2DM liability consistently attenuated on adjustment for genetically-predicted BMI, although the attenuation was not substantial. Estimates for HbA1c were largely unchanged following adjustment for genetically-predicted BMI. This suggests that a pleiotropic or mediating effect of BMI between T2DM variants and cardiovascular diseases contributed to the estimates of T2DM in primary analyses, whereas there was no such contribution to estimates for HbA1c (Appendix A).

## 4. Discussion

In this study, we employed MR to investigate potential impact of T2DM liability and elevated HbA1c on a wide range of cardiovascular disease outcomes in population-based cohort study. Our findings reinforce the importance of optimizing glycaemic control for the prevention of a broad range of cardiovascular disease outcomes. The attenuation of MR effect estimates in individuals without diabetes suggests that the benefit of glycaemic control is greatest in diabetic patients.

This study builds on previous epidemiological evidence. Our findings are consistent with previous MR studies supporting effects of T2DM liability on CAD and ischemic stroke [3,4]. We here provide genetic evidence supporting additional effects of T2DM liability on peripheral vascular disease, heart failure and aortic valve stenosis. Our findings are also consistent with previous cohort studies [12], but are additionally less prone to environmental confounding. We replicated previous evidence for causal effects of hyperglycemia on CAD and stroke [13]. Interestingly, we did not identify an association between genetically-predicted HbA1c and ischemic stroke, which has been observed in both observational and genetic studies [2,4]. This may possibly be because of insufficient statistical power in our current analysis.

Of relevance, evidence from clinical trials does not support intensive glycaemic control for reducing the risk of combined major macrovascular disease or subtypes such as CAD and stroke [14,15]. This discrepancy may relate to the mode of glucose lowering, as well as its intensity—previously considered pharmacological interventions for glycaemic control might exert adverse effects that could negate the benefit of improved glycaemic control. Another potential reason is that genetic associations represent lifelong differences in average blood glucose levels, while trials can only evaluate the impact of short-term interventions.

Our previous MR study in diabetes-free individuals (without diabetes and baseline HbA1c < 6.5%) of European ancestries supported a dose-response relationship between genetically-predicted HbA1c and incident CAD risk with no evidence to reject the null hypothesis of a constant linear effect at different values of HbA1c [5]. Our current study indicates that the associations of genetically-predicted HbA1c with CAD risk for individuals in the normoglycemic and prediabetic range are attenuated as compared to when also considering diabetics. This discrepancy compared with our previous study was unexpected, and may arise because of the different case definitions: in the previous study, we only investigated incident CAD, while here we further included prevalent CAD events. Another possible reason is the different criteria for variant selection: in the previous study, we performed a single analysis including variants associated with both T2DM and HbA1c, whereas here we included variants associated with either trait in separate analyses. The current strategy resulted in a much larger number of genetic variants being included in the analysis, which is generally regarded as positive, but here it may have diluted estimates by increasing the between-variant heterogeneity.

A strength of our study is that we used MR approaches which are robust to the impact of confounding and reverse causation. We also strengthened our findings by performing sensitivity analyses to assess the consistency of MR estimates using robust methods that make different assumptions about the genetic variants. Another strength is that we simultaneously assessed the associations with different outcomes comprehensively covering most macrovascular complications. Thereby we were able to compare the directions and sizes of associations and investigate differences across disease outcomes.

Our study has several limitations. We selected UK Biobank participants of European ancestries to control the impact of population stratification. However, evidence from this study may not be generalizable to other ethnic groups. The substantial between-variant heterogeneity in the primary analyses suggests potential bias due to genetic pleiotropy across different genetic variants. However, estimates were broadly similar in robust methods that make different assumptions with respect to pleiotropy. Due to the low statistical power for some uncommon outcomes, non-significant estimates for these outcomes do not provide strong evidence supporting the absence of associations. Data from UK Biobank for selecting and estimating the associations of variants of HbA1c were largely overlapping with the analytic sample for genetic associations with the outcome. While the F-statistic for HbA1c genetic variants suggests a moderate strength of the instruments and hence limited bias that would be unlikely to affect our conclusions, we cannot completely exclude possible bias from weak instruments or winner’s curse. We avoided conducting analyses restricted to only patients with diabetes, because the management of diabetes in this group (for example, use of antiglycemic medications) would affect the validity of the genetic variants for predicting glycemic control, potentially rendering the genetic instruments invalid and making findings difficult to interpret.

## 5. Conclusions

In conclusion, this wide-angled MR study provides genetic evidence supporting effects of T2DM liability and glycemic control on the risk of a wide range of cardiovascular disease outcomes. Lowering average blood glucose level is likely to reduce cardiovascular risk, with the strongest effects in individuals with diabetes.

## Figures and Tables

**Figure 1 genes-12-01644-f001:**
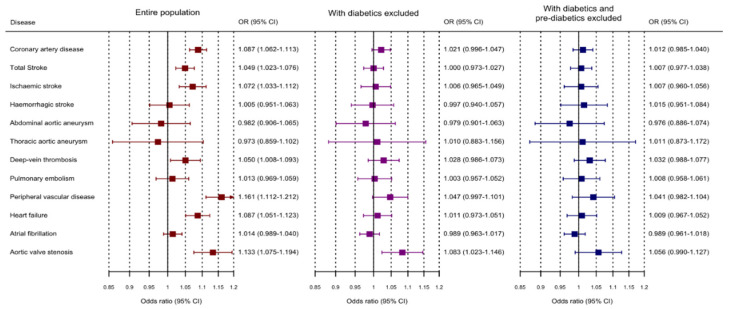
Mendelian randomization estimates (odds ratios with 95% confidence intervals) for cardiovascular outcomes per 2-fold increase in genetically predicted risk of type 2 diabetes mellitus. Analyses were performed in 367,703 UK Biobank participants of European ancestries, and in subsets of participants without diabetes, and participants without diabetes or pre-diabetes.

**Figure 2 genes-12-01644-f002:**
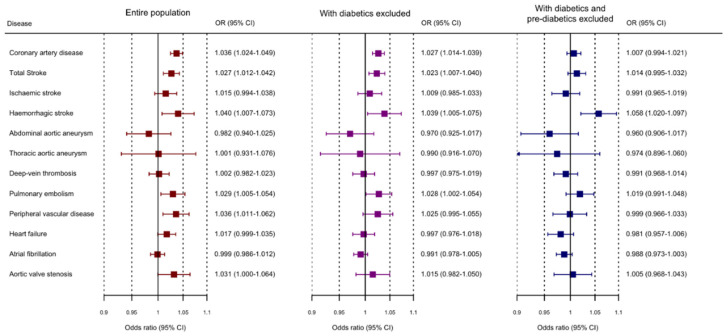
Mendelian randomization estimates (odds ratios with 95% confidence intervals) for cardiovascular outcomes per 1 mmol/mol increase in genetically predicted HbA1c levels. Analyses were performed in 367,703 UK Biobank participants of European ancestries, and in subsets of participants without diabetes, and participants without diabetes or pre-diabetes.

**Table 1 genes-12-01644-t001:** Baseline characteristics of UK Biobank participants included in the present study.

Baseline Characteristics	All Participants	With Diabetics Excluded	With Diabetics and Pre-Diabetics Excluded
Number of Participants	367,703	330,825	284,740
Female (%)	198,860 (54.1%)	182,200 (55.1%)	156,875 (55.1%)
Mean age at baseline (SD), years	57.2 (8.0)	57.0 (8.1)	56.4 (8.1)
Body mass index (SD), kg/m^2^	27.4 (4.8)	27.1 (4.6)	26.8 (4.4)
HbA1c (SD), mmol/mol	35.5 (6.6)	34.4 (3.3)	34.4 (3.3)
Systolic blood pressure (SD), mmHg	137.7 (18.6)	137.4 (18.7)	136.8 (18.6)
Diastolic blood pressure (SD), mmHg	82.0 (10.1)	82.0 (10.1)	81.8 (10.1)
Smoking status (%)			
Current	37,860 (10.3%)	33,891 (10.2%)	26,760 (9.4%)
Ex	185,668 (50.5%)	166,321 (50.3%)	143,469 (50.4%)
Never	143,749 (39.1%)	130,511 (39.5%)	114,430 (40.2%)
Alcohol consumption status (%)			
Current	342,733 (93.2%)	309,764 (93.6%)	267,779 (94.0%)
Ex	12,729 (3.5%)	10,727 (3.2%)	8642 (3.0%)
Never	11,642 (3.2%)	10,100 (3.1%)	8129 (2.9%)

Baseline characteristics are presented as mean (standard deviation, SD) or *n* (%). Participants with missing information for the given measurement were not included in the calculation of mean and standard deviation, and were omitted from the categorization by smoking and alcohol status.

## Data Availability

The genetic variants used as instrumental variables are detailed in Appendix A.

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
