# Peer review of "Genetically Predicted Type 2 Diabetes Mellitus Liability, Glycated Hemoglobin and Cardiovascular Diseases: A Wide-Angled Mendelian Randomization Study"

_genes, 2021, doi:10.3390/genes12101644_

Round 1
Reviewer 1 Report
Summary
The study from Bowen Liu et al. describes two sample mendelian randomization analyses based on summary-level data involving 243 variants as genetic instruments for T2DM liability and 536 variants for HbA1c. The authors show positive associations of increased genetically predicted T2DM liability and HbA1c levels with higher risk of several cardiovascular diseases. They used data from large consortium and UK Biobank case control results. The scope of the study is clear, and the manuscript is well written. The statistical methods are straightforward and described in detail.
Major comments
- I thank the authors for describing in detail how genetic instruments for HbA1c were selected from a genome-wide association study in 408,989 UK Biobank participants of European ancestries and without diabetes. In discussion, they acknowledged this as a limitation because the sample overlapping issue explained in the following “data from UK Biobank for selecting and estimating the associations of variants of HbA1c were largely overlapping with the analytic sample for genetic associations with the outcome”. Which is a source of bias from weak instruments or winner’s curse.
Why the authors did not use an external summary statistics (like for T2DM) to select genetic instruments for HbA1c? For example, in GWAS catalog many summary statistics are available from European based GWAS of HbA1c. At least such external summary statistics of hbA1c could be used for validation analysis.
- I appreciate the fact that the authors conducted complementary and sensitivity analyses using robust MR methods. However, describing those results was limited. In results section, it is stated that estimates from weighted median and MR-Egger were similar to those obtained in the principal analysis using IVW without any comment on p-values which were not significant for all outcomes except for CAD when using weighted median. Additionally, MR-Egger intercept showed evidence for pleiotropy.
The authors are invited to describe sensitivity analysis and discuss more clearly the differences obtained using the different methods.
- In the conclusion, the authors stated that “Lowering average blood glucose level is likely to reduce cardiovascular risk, with the strongest effects in individuals with diabetes”. However, analyses were conducted in all participants from UK biobank cohort and in two subgroups, (1) excluding participants with diabetics and (2) excluding participants with diabetics and pre-diabetics.
What about analyzing the associations in group of diabetics only to confirm the conclusion?
Minor comments
- I thank the authors for the well explained used methods. However, it seems that some details were lacking or confusing. For example,
- in lines 91-92 they describe identifying 243 variants associated with T2DM from DIAGRAM consortium GWAS the main criteria used was the genome-wide significance level of p<5x10-8 but they don’t state which LD r2 threshold was retained for clumping or pruning.
- In abstract (lines 25-27) and in methods section (lines 94-95) they gave definition of diabetes criteria when talking about participants without diabetes. It will be less confusing if definitions were concordant and straightforward as used in lines 112-114.
Reviewer 2 Report
The abstract is constructive and up to the point.
The aim and research gap was clearly explained. The background information was adequate.
Materials and methods were clearly explained.
The results and conclusions have adequate explanation. Table and figures were relevant and clearly explained.
The discussion explained the significance of the results and short comings.
